# Molecular Detection of Porcine Cytomegalovirus, Porcine Parvovirus, Aujeszky Disease Virus and Porcine Reproductive and Respiratory Syndrome Virus in Wild Boars Hunted in Serbia during 2023

**DOI:** 10.3390/vetsci11060249

**Published:** 2024-06-03

**Authors:** Nemanja Jezdimirović, Božidar Savić, Bojan Milovanović, Dimitrije Glišić, Milan Ninković, Jasna Kureljušić, Jelena Maletić, Jelena Aleksić Radojković, Dragan Kasagić, Vesna Milićević

**Affiliations:** 1Scientific Institute of Veterinary Medicine of Serbia, Janisa Janulisa 14, 11000 Belgrade, Serbia; 2Forensic Veterinary Medicine, Faculty of Veterinary Medicine, University of Belgrade, Bulevar Oslobođenja 18, 11000 Belgrade, Serbia; 3PI Veterinary Institute of the Republic of Srpska “Dr. Vaso Butozan” Banja Luka, Branka Radičevića 18, 78000 Banja Luka, Republic of Srpska, Bosnia and Herzegovina

**Keywords:** PCMV, ADV, PPV, PRRS, wild boar, real-time PCR

## Abstract

**Simple Summary:**

This study focused on investigating the presence of porcine cytomegalovirus (PCMV) and other viral infections in Serbian wild boars. Using real-time PCR, samples from 50 wild boars were analyzed for PCMV, Aujeszky disease virus (ADV), Porcine parvovirus (PPV), and Porcine reproductive respiratory syndrome virus (PRRSV). Results revealed an 8% PCMV infection rate, with females showing higher susceptibility. PPV was detected in 56% of samples and ADV in 18%, while PRRSV was not found. These findings underscore the zoonotic potential of PCMV and highlight wild boars as reservoirs of various pathogens, posing risks to both the pig industry and public health.

**Abstract:**

Porcine cytomegalovirus (PCMV) infection is widespread worldwide and has a high prevalence in swine herds, especially in countries with intensive swine production. PCMV is zoonotic and can impact xenotransplants. It is the third swine virus known to be zoonotic, following swine influenza virus (influenza A) and hepatitis E virus genotype 3 (HEVgt3 or HEV-3). Wild boars, serving as reservoirs for various pathogens, including PCMV, pose a risk to both the pig industry and public health. This study aimed to investigate PCMV infection in Serbian wild boars using real-time PCR and assess other viral infections. We also tested samples for the presence of other viral infections: Aujeszky disease virus (ADV), Porcine parvovirus (PPV) and Porcine reproductive respiratory syndrome (PRRSV). Samples from 50 wild boars across 3 districts were tested. Results showed 8% positivity for PCMV DNA, with females showing higher infection rates. Porcine parvovirus (PPV) was detected in 56% of samples, while Porcine reproductive respiratory syndrome virus (PRRSV) was absent. ADV was found in 18% of samples, primarily in younger animals. This research contributes to understanding PCMV prevalence in Serbian wild boars and emphasizes the importance of monitoring viral infections in wild populations, considering the potential zoonotic and economic implications.

## 1. Introduction

Porcine cytomegalovirus (PCMV) infection is widespread worldwide and has a high prevalence in swine herds, especially in countries with intensive swine production [1]. It often goes unnoticed clinically due to immunity acquired early in life. Porcine cytomegalovirus, despite its misleading name, belongs to the *Betaherpesvirinae* subfamily and the *Herpesviridae* family [2]. The name PCMV likely originated from the presence of cytomegalic cells with characteristic basophilic intranuclear inclusion bodies in the mucosal glands of pig turbinates [3]. PCM virus is not assigned to any specific genus. Genetic research indicates that PCMV is more closely related to the human herpesviruses 6A, 6B, and 7 (HHV-6A, -6B, and -7) than to human cytomegalovirus (HCMV, also called human herpesvirus 5, HHV-5) [4]. It is the third swine virus known to be zoonotic, following swine influenza virus (influenza A), hepatitis E virus genotype 3 (HEVgt3 or HEV-3), and has been associated with significantly decreased survival times for xenotransplants of pig kidneys or hearts in non-human primates [5]. Wild boars can act as reservoirs for various bacteria, viruses, and parasites that can be transmitted to humans and domestic animals through direct contact, contaminated food, or environmental contamination [6]. Wild boars can act as reservoirs for PCMV; thus, free-ranging wild boar populations are increasingly considered to be a threat to the pig industry, as well as public health because of the potential effects of PCMV transmission in preclinical xenotransplantation [7,8,9]. Contrary to the established prevalence of PCMV in many countries in domestic pigs, the prevalence of PCMV in wild boar populations is poorly investigated. The first detection of PCMV in a wild boar occurred in Japan in 2013 [10] and it has also been identified in wild boars in Italy, Germany [5], Russia, and Argentina [1,11]. However, in contrast, the PCMV infection in wild boars is rarely investigated, though there are reports on its occurrence. In the Republic of Serbia, PCMV infection in domestic pigs was reported for the first time in 2023 [12]. Despite the widespread presence of PRRSV in domestic pigs, there is limited knowledge about PRRSV infection in European wild boars (Sus scrofa). In Lithuanian wild boar populations, PRRSV was found to be highly prevalent, with approximately 18.66% detected using conventional RT-PCR and 19.54% using real-time RT-PCR [13]. Earlier serological studies conducted on wild boars in Serbia were restricted in scope but indicated a comparatively high prevalence of AD antibodies. Lazić et al., [14] documented an average prevalence of 38.21%, with the highest percentage observed in populations older than 2.5 years, reaching 46.86%. ADV seroprevalence in different regions varies, with the highest percentage found in the east, with an average of 83% [15]. Detection and sequence analysis of PPVs in wild boar population in Serbia revealed the presence of PPV1, PPV2, and PPV3 DNA in 56.6% of the samples analyzed, with PPV4 not being detected. Among the positive samples, PPV3 was the most prevalent, appearing in 69.6% of cases, followed by PPV1 in 63.8%, and PPV2 in 21.7% of samples ([16]). Thus, the aim of this study was to complement the knowledge about PCMV infection in Serbia by testing samples from wild boars for the presence of PCMV DNA using real-time PCR. In addition, we also tested samples for the presence of other viral infections: Aujeszky disease virus (ADV), Porcine parvovirus 1 (PPV), and Porcine reproductive respiratory syndrome (PRRSV).

## 2. Materials and Methods

### 2.1. Study Design and Sampling

The tissue samples originated from free-living wild boars hunted during the 2023 hunting season in the territory of the Republic of Serbia, specifically from three administrative districts: Južnobanatski, Borski and Zaječarski districts (Figure 1).

In total, 50 tissues samples (lymph nodes, spleen, and kidney) were collected. The age of the shot wild boars was estimated based on dentition characteristics, following the recommendations of SC (EDA SCHEDA Ecological Associates, Inc.) as outlined below: 0–6 months old: 0 permanent molars; 6–18 months of age: 1 permanent molar; 1.5–2.5 years of age: 2 permanent molars; older than 2.5 years: 3 permanent molars. The estimated age range varied from 6 to 30 months. After that age, dentition is no longer an accurate indicator of an animal’s age, and therefore, pigs older than 30 months were classified in a single category.

### 2.2. Molecular Detection of DNA

The extraction of pulled tissue samples containing lymph nodes, spleen, and kidney was homogenized and samples were prepared as 1:10 suspension in phosphate buffered saline (PBS; pH 7.2). The suspension was centrifuged for 10 min at 2000× *g* and RNA Nucleic acid was extracted from the supernatant using IndiSpin Pathogen kit (Indical, Leipzig, Germany) according to the manufacturer’s instructions. Real-time PCR for PCMV, PPV and ADV detection and real-time RT-PCR for PRRSV ORF7 gene were performed using previously published primers in Table 1 [17,18,19,20] and commercial kits Luna Universal qPCR Master Mix (NEB, Ipswich, MA, USA) and Luna^®^ Universal One-Step RT-qPCR Kit (NEB, Ipswich, MA, USA). An external VetMax Xeno Internal positive DNA/RNA control (Applied Biosystems, Thermo Fisher Scientific, Waltham, MA, USA) was included in each sample. The samples with Ct values of ≤40 were considered positive.

### 2.3. Statistical Analysis

The obtained results were analyzed using descriptive statistics. The statistical analyses were performed using JASP (JASP Team, version 0.16.0).

## 3. Results

Samples of 50 wild boars’ organs from the Južnobanatski, Borski, and Zaječarski districts were classified according to gender and age. The animals are divided into two categories: (1) up to 18 months of age (38 pigs) and (2) older than 19 months (12 pigs). The body weight of the animals ranged from 11 to 96 kg. In total, 50 wild boars were examined, 22 were male (44%) and 28 were female (56%). According to age, most wild boars belonged to the group 6–18 months of age (n = 38, 76%) The results are shown in Table 2.

Out of 50 tissues samples from wild boars, 4 were positive for the presence of PCMV DNA (n = 4, 8%). Two affirmative samples were obtained from Južnobanatski district, with one originating from a male and another from a female. Additionally, two samples tested positive in the Borski district, both of which were from female boars. Table 3 displays the results.

Porcine parvovirus (PPV) was positive in 28 out of the 50 examined animals (28/50, 56%). A total of 17 of the 28 positive samples were obtained from female animals 17/28 (60%).

The Borski district had the highest number of positive cases. The results are shown in Table 4.

The PRRS virus was not identified in any of the samples examined.

ADV genome was detected in 9 of 50 samples (18%). All ADV positive (n = 9) wild boars belonged to the group of 6–18 months old wild boar (females, n = 5; males, n = 4). The results are shown in Table 5.

## 4. Discussion

Wild boars are recognized as reservoirs for various diseases. In the Republic of Serbia, there is sparse information regarding the impact of wild boars on the prevalence of infectious diseases in domestic pigs, and data on the occurrence and spread of diseases among wild boars is limited on certain viral diseases. In addition to the documented seroprevalence of classical swine fever, Bovine viral diarrhea virus in wild boar, Porcine kobuvirus [15,21,22,23], hepatitis E [24], Aujeszky’s disease [15,25], Porcine parvovirus 1 [16], and porcine circovirus type 2 (PCV2) [26], there are no other indicators available regarding the health status of the wild boar population in the Republic of Serbia. In order to obtain representative results from this study, wild boar sampling was conducted in three regions, including the hunting area with the highest density of wild boars in the Borski district (five wild boar per km^2^) [27], situated in the Timok-Krajina region. Additionally, obtaining information about the age and gender of the hunted wild boars might be important to understand the timing and dynamics of disease occurrence in preceding years. Demographic factors such as gender and age were examined concerning PCMV status. The analysis revealed that the infection rate was higher in females, accounting for 10.71%, whereas it was 4.54% in males. Contrary to these results, De Maio et al. [1] claimed that the proportion of infected individuals was very similar between sexes, accounting for 52.4% in females and 53.6% in males. Testing of young animals (aged between 6 and 18 months) was carried out with the goal of collecting data about the presence of different viruses in wild boar populations, as young animals are more susceptible to testing positive for certain viruses compared to older animals that might have developed immunity. Therefore, 76% of the tested animals from our study belonged to this target age group. PCMV infection typically occurs in younger animals, primarily through the transmission of the viral infection from the wild boar to the wild boar piglets, especially post-farrowing [3]. The lower occurrence of infection in older pig categories suggests that direct animal contact does not necessarily result in an increased percentage of infected animals. However, it cannot be conclusively ruled out that the antibody titer in infected animals decreases over time or that the animal might succumb to the clinical disease [28]. The initial instance of a wild boar with cytomegalic cells and virologically confirmed PCMV infection was documented in Japan [10]. In Northeastern Patagonia, Argentina (Buenos Aires and Río Negro Provinces), a screening of PCMV using a nested PCR assay on tonsil tissues from 62 freely roaming wild boars revealed an overall infection rate of approximately 56% [1]. In the same report, a notably higher prevalence of nearly 90% was identified in animals aged less than 6 months. In our research, we established the presence of PCMV in 10.53% of wild boars belonging to the group that was 6–18 months of age. Based on the available reports, it has been proven that detection of the PCMV genome using real-time PCR is significantly more frequent in wild boars up to 22 months of age compared to older categories, where the infection is most likely present in a latent form [5]. In a report by Hansen et al. [5], utilizing the Western blot method, a higher percentage of PCMV detection was observed in pigs less than 22 months of age (46%) compared to older categories (44%). The results obtained by using real-time PCR were similar to the detection rates of the overall population (46% by Western blot and 30% by PCR). In animals more than 22 months old, no differences were observed in the detection rate by Western blot and PCR (44% in both cases). A probable explanation for the higher detection rate of PCMV infection in older animals by Western blotting may be that the latent state of infection with minimal shedding to the environment does not conform to the distribution pattern of infection as seen in young animals, making it more frequently detected using real-time PCR. Furthermore, during latent infection, PCMV is restricted to specific organs, including the lung, liver, salivary gland, and kidney [3,7], and the frequency of infection reactivation remains unknown. Therefore, further research is essential to achieve a deeper understanding of this phenomenon [3]. There are indications that the majority of infections occur in younger animals, with potential transmission between mothers and piglets [29]. In our research, 56% of the examined samples tested positive for the presence of porcine parvoviruses (PPV), a result comparable to a study conducted in the USA which showed infection with PPV type 1 [30] in wild boars. Additionally, in the research of Nišavić et al. [16], the detection rate of the DNA of PPV1, PPV2, and PPV3 (56.6%) was similar to that of our study, while the authors report that they did not identify PPV4. According to same authors, on PPV identification in wild boar populations, the estimated prevalence of PPV genotypes is 69.6% for PPV3, 63.8% for PPV1, and 21.7% for PPV2. There were no positive samples for PRRSV in our investigation, which aligns with the findings of Kukushkin et al. [31], who also did not detect PRRSV-positive animals in wild boar populations across five regions in Russia. Considering that we utilized samples of lymph nodes, spleen, and kidney, we cannot entirely rule out the possibility of PRRS presence in the wild boar population in Serbia. However, other researchers have reported PRRSV in wild boar populations from various European countries [32,33], including Belarus and the Kaliningrad Region [13]. Therefore, it was concluded that wild boars may serve as a natural reservoir for PRRSV. Consequently, PRRSV is highly prevalent in Lithuanian wild boar populations, with an average prevalence rate of 18.66% established by conventional RT-PCR and 19.54% using real-time PCR [13]. Aujeszky’s disease, caused by *Suid herpesvirus* 1, is a viral ailment affecting domestic and wild suids and has a global presence, exerting a substantial economic impact. In the Republic of Serbia, there is no established program for eradicating Aujeszky’s disease or implementing a national vaccination regimen for domestic pigs [15]. The Aujeszky’s disease impacts the reproductive and health conditions of wild boars, especially in piglets, and the increased presence of the viral genome in piglets and genital swabs suggests transmission through both vertical and venereal routes [34]. The ADV genome was identified in (9/50, 18%), specifically in wild boars aged between 6 and 18 months (females, n  =  5; males, n  =  4). Our findings align with those reported by Milićević et al. [15]. Numerous studies have suggested a higher incidence of Aujeszky’s disease in females, attributed to their behavioral patterns [26,35,36]. Females, residing in groups with more frequent animal contacts, are deemed to be at a heightened risk of ADV infection. Moreover, older and more socially active adults face an increased risk of infection [37].

## 5. Conclusions

The detection of the PCMV genome in the tissue sample of wild boar indicated the circulation of the virus in wild boar populations in Republic of Serbia. These results indicate a relatively low but not negligible prevalence of the virus in the wild boar population, which may represent a significant risk for the domestic pig population as well. A total of 56% of the examined samples were positive for the presence of porcine parvoviruses (PPV), the results of our investigation of PPV also show high circulation of the virus in wild boar population. Wild boars may act as a natural reservoir for PRRSV, but in our investigation there was no presence of the PRRSV genome in wild boars. Wild boars excreting ADV transmit it to domestic pigs, which is why infections in extensive production become feasible. It is crucial to take into account the potential interactions between pigs and wild boars when working towards the eradication of ADV.

## Figures and Tables

**Figure 1 vetsci-11-00249-f001:**
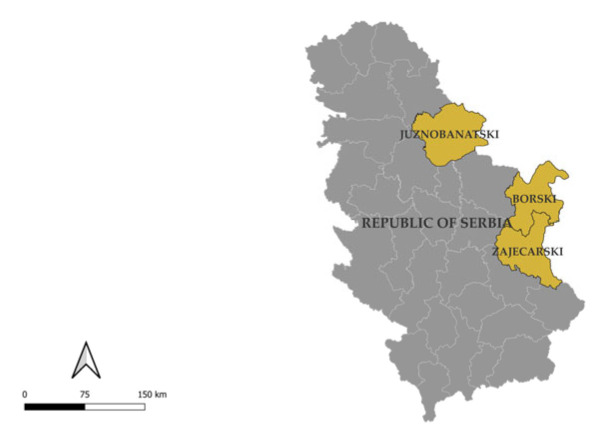
Administrative districts in Serbia where the tested samples were collected.

**Table 1 vetsci-11-00249-t001:** Primers used in the study to detect the RNA/DNA of PCMV, PPV, ADV, and PRRS.

Name	Primer Sequence	Reference
**PCMV-F**	3′-GCTGCCGTGTCTCCCTCTAG-5′	[17]
**PCMV-R**	3′-ATTGTTGATAAAGTCACTCGTCTGC-5′	
**PCMV-P**	TAMRA-CCATCACCAGCATAGGGCGGGAC-FAM	
**PRRS-ORF7QF**	3′-GCTGAAGATGACRTYCGGCA-5′	[18]
**PRRS-ORF7QR**	3′-GCAGTYCCTGCGCCTTGAT-5′	
**PRRS-ORF7QP**	TAMRA -TGCAATCGATYCAGAC-FAM	
**PPV-F**	3′-CCAAAAATGCAAACCCCAATA-5′	[19]
**PPV-R**	3′-TCTGGCGGTGTTGGAGTTAAG-5′	
**PPV-P**	TAMRA-CTTGGAGCCGTGGAGCGAGCC-FAM	
**ADV gB718F**	3′-ACAAGTTCAAGGCCCACATCTAC-5′	[20]
**ADV gB812R**	3′-GTCYGTGAAGCGGTTCGTGAT-5′	
**ADV gB785P**	TAMRA -ACGTCATCGTCACGACC-FAM	

**Table 2 vetsci-11-00249-t002:** Wild boars classified according to gender and age.

District	Gender		Age	
	Male (%)	Female (%)	6–18 Months	>19 Months
**Južnobanatski**	10/15 (66.6%)	5/15 (33.3%)	14 (93.3%)	1 (6.7%)
**Borski**	6/23 (26.1%)	17/23 (73.9%)	14 (60.8%)	9 (39.2%)
**Zaječarski**	6/12 (50%)	6/12 (50%)	10 (83.3%)	2 (16.7%)
**Total**	**22/50 (44%)**	**28/50 (56%)**	**38/50 (76%)**	**12/50 (24%)**

**Table 3 vetsci-11-00249-t003:** Results positive for the presence of DNA of porcine cytomegalovirus.

District	PCMV	Gender		Age	
		Male (%) (n = 22)	Female (%) (n = 28)	6–18 Months	>19 Months
	Total 4/50 (8%)	1/22 (4.54%)	3/28 (10.71%)	4/38 (10.53%)	0/12 (0%)
**Južnobanatski ** **(n = 15)**	2/15 (13.3%)	1/15 (6.6%)	1/15 (6.6%)	2/14 (14.3%)	0/1 (0%)
**Borski** **(n = 23)**	2/23 (8.7%)	0/23 (0%)	2/23 (8.7%)	2/14 (14.3%)	0/9 (0%)
**Zaječarski ** **(n = 12)**	0/12 (0%)	0/12 (0%)	0/12 (0%)	0/10 (0%)	0/2 (0%)

**Table 4 vetsci-11-00249-t004:** Results of positive presences of DNA of Porcine parvovirus.

District	PPV	Gender		Age	
		Male (%) (n = 22)	Female (%) (n = 28)	6–18 Months	>19 Months
	Total 28/50 (56%)	11/22 (50%)	17/28 (60.7%)	19/38 (50%)	9/12 (75%)
**Južnobanatski** **(n = 15)**	7/15 (46.6%)	2/15 (13.3%)	5/15 (33.3%)	7/14 (50%)	0/1 (0%)
**Borski** **(n = 23)**	17/23 (73.9%)	8/23 (34.8%)	9/23 (39.1%)	8/14 (57.1%)	9/9 (100%)
**Zaječarski** **(n = 12)**	4/12 (33.3%)	1/12 (8.3%)	3/12 (33.3%)	4/10 (40%)	0/2 (0%)

**Table 5 vetsci-11-00249-t005:** Results of positive presence of DNA of ADV in organs of wild boars.

District	ADV	Gender		Age	
		Male (%) (n = 22)	Female (%) (n = 28)	6–18 Months	>19 Months
	Total 9/50 (18%)	4/22 (18.18%)	5/28 (17.86%)	9/38 (23.7%)	0/12 (0%)
**Južnobanatski** **(n = 15)**	3/15 (20%)	0/15 (0%)	3/15 (20%)	3/14 (21.4%)	0/1 (0%)
**Borski** **(n = 23)**	4/23 (17.4%)	3/23 (13.04%)	1/23 (4.3%)	4/14 (28.6%)	0/9 (0%)
**Zaječarski** **(n = 12)**	2/12 (16.6%)	1/12 (8.3%)	1/12 (8.3%)	2/10 (20%)	0/2 (0%)

## Data Availability

All data are contained within the manuscript.

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
