# Peer review of "Molecular Detection of Porcine Cytomegalovirus, Porcine Parvovirus, Aujeszky Disease Virus and Porcine Reproductive and Respiratory Syndrome Virus in Wild Boars Hunted in Serbia during 2023"

_vetsci, 2024, doi:10.3390/vetsci11060249_

Round 1

Reviewer 1 Report

Comments and Suggestions for Authors

The ms entitled “Molecular detection of Porcine cytomegalovirus, Porcine parvovirus, Aujeszky disease virus and Porcine reproductive and respiratory syndrome virus in wild boars hunted in Serbia during 2023” showed the prevalence of Porcine cytomegalovirus, Porcine parvovirus, Aujeszky disease virus and Porcine reproductive and respiratory syndrome virus in wild boars in Serbia during 2023. The author underscored the zoonotic potential of PCMV and highlight wild boars as reservoirs of various pathogens. The authors undertook an interesting work, and the results provides more insight for the prevention and control of porcine disease. However, there are some clerical errors in the ms, please check and correct.

Some comments are list below:

1.        In the introduction part, it is preferred to give a brief introduction on the on the epidemic conditiond of the other three virus, Aujeszky disease virus (ADV), Porcine parvovirus 1 (PPV1) and Porcine reproductive respiratory syndrome (PRRSV) in domestic and wild pigs worldwide including Serbia.

2.        The “2. Materials and Methods” part should be listed in form of numbering, such as 2.1 Ethical review and approval……, 2.2 sample collection……

3.        Line 79, figure legend should be placed below the figure.

4.        Line 89, what does “PBS 7.2” mean?

5.        Line 90, “The suspension was centrifuged for 10 min at 2000 g and RNA.”, please check this sentence.

6.        Primers used in this study should be listed in manuscript, such as  in table.

7.        The software used for statistical analysis should be clarified.

8.        Line 112-113, “Table number 2” to Table 2.

9.        Table 3,  in the forst row, the sum of positive cases in the fifth and sixth columns is greater than the total positive cases (28). In the forth row, the number of cases is also inconsistent. The problem is also exist in Table 4.

10.    Line 129, according to Table 5, females, 5; males, 4. Please check which is correct.

11.    Please provide information on the source organ of the positive samples.

Author Response

Dear reviewer 1 plesae see the attachment. 

Best regards. 

Reviewer 2 Report

Comments and Suggestions for Authors

The manuscript describes work which contributes to the body of knowledge regarding porcine viruses in wild boar. The introduction covers the prior knowledge in the field and the work of others is cited comprehensively. The methods are sufficiently detailed for the work to be repeated by others if necessary and the results and discussion seem to be sound. I have no hesitation in recommending publication following an english spell-check. 

Comments on the Quality of English Language

Please see above. 

Author Response

Dear Reviewer 2 please see the attachment.

Best regards.

Reviewer 3 Report

Comments and Suggestions for Authors

In this manuscript, Jezdimirović N et al. studied the molecular detection of Porcine cytomegalovirus, Porcine parvo-2 virus, Aujeszky disease virus, and Porcine reproductive and respiratory syndrome virus in wild boars. The study is potentially interesting to readers. The manuscript is straightforward, but some of the data are not strictly presented and the controls are missing. Here are some specific suggestions to consider:

Main comments:

1. Table 4. Results of positive presence of PRRS virus in wild boars. The author claimed 0% of the samples were positive, however, in Table 4, they showed 66.6% of males were positive and so on. It confused the readers which is correct. Strongly recommend the author repeat the experiments.

2. Overall, the cell control of qPCR was missing. Ex. GAPDH mRNA as a control. The relative CT value should be shown in support material.

3. The author didn’t show which tissue they detected. I think they should detect different viruses by using different tissues. Ex. PRRSV should use lung. Perhaps that is why they cannot detect PRRSV.

Author Response

Dear Reviewer 3 please see the attachment.

Best regards.

Round 2

Reviewer 3 Report

Comments and Suggestions for Authors

The author argued, "The samples with Ct values of ≤40 were considered positive." It should be based on the negative control, eg, non-infected tissue.

Figure 1 is unreadable. It seems from the internet which should be added the resource.

The full name and abbreviation of Aujeszky disease virus (ADV), Porcine parvovirus 1 (PPV1) showed up in the text again and again.

Line 84: hinting to Hinting.

Comments on the Quality of English Language

no

Author Response

Dear Reviewer 3,

Attached please find the answers to your comments.

Best regards
